# Partial Venous Inflow Occlusion under Mild Hypothermia for Membranectomy in a Dog with Cor Triatriatum Dexter

**DOI:** 10.3390/ani13182921

**Published:** 2023-09-14

**Authors:** Chang-Hwan Moon, Kyung-Min Kim, Won-Jong Lee, Woo-Jin Kim, Seok-Min Lee, Ho-Jung Choi, Hae-Beom Lee, Seong-Mok Jeong, Dae-Hyun Kim

**Affiliations:** 1Department of Veterinary Surgery, College of Veterinary Medicine, Chungnam National University, 99, Daehak-ro, Yuseong-gu, Daejeon 34134, Republic of Korea; moonch0208@gmail.com (C.-H.M.); kgmtong8886@daum.net (K.-M.K.); wjl03ssaaa@naver.com (W.-J.L.); loneless1175@naver.com (W.-J.K.); seatiger76@cnu.ac.kr (H.-B.L.); jsmok@cnu.ac.kr (S.-M.J.); 2Department of Veterinary Medical Imaging, College of Veterinary Medicine, Chungnam National University, 99, Daehak-ro, Yuseong-gu, Daejeon 34134, Republic of Korea; seokmin1992@gmail.com (S.-M.L.); hjchoi@cnu.ac.kr (H.-J.C.)

**Keywords:** cor triatriatum dexter, partial venous inflow occlusion, mild hypothermia, membranectomy, cardiac surgery, dog

## Abstract

**Simple Summary:**

This report describes a case of cor triatriatum dexter in which a persistent embryonic membrane within the right atrium obstructed venous blood inflow from the caudal vena cava. Several treatment options for dogs presenting with clinical signs related to cor triatriatum dexter, such as interventional approaches, cardiopulmonary bypass, or venous inflow occlusion, have been published in the veterinary literature. Each method has advantages and disadvantages; however, membranectomy via venous inflow occlusion is considered an effective and safe treatment compared with other methods, as it can be performed within a short period and reduces the risk of recurrence. To the best of our knowledge, this is the first case report documenting the surgical treatment of a cor triatriatum dexter in a dog using partial venous inflow occlusion under mild hypothermia.

**Abstract:**

Cor triatriatum dexter (CTD) is an uncommon congenital cardiac anomaly in dogs. This case report describes successful membranectomy for CTD via partial venous inflow occlusion under mild hypothermia in a dog. A 7-month-old intact male mixed-breed dog weighing 20.5 kg presented with a history of abdominal distention, lethargy, and anorexia. Clinical examination, radiography, echocardiography, microbubble testing, and computed tomography revealed a remnant right atrium membrane obscuring the venous blood inflow from the vena cava. Considering the potential risk of re-stenosis following interventional treatment, curative resection involving surgical membranectomy via venous inflow occlusion was performed. By performing partial venous inflow occlusion under mild hypothermia (34.5 °C), sufficient time was obtained to explore the defect and resect the remnant membrane. The dog recovered without any complications, and the clinical signs were relieved. This case illustrates that partial venous inflow occlusion under mild hypothermia is feasible for achieving curative resection of cor triatriatum dexter in dogs.

## 1. Introduction

Cor triatriatum dexter (CTD) is a rare obstructive congenital cardiovascular malformation in dogs [1]. CTD is an embryogenic malformation that occurs when the right sinus venosus valve, which normally regresses during embryogenesis, remains, resulting in double chambers in the right atrium [2]. The persistent embryonic membrane in the right atrium interferes with the venous blood inflow from the caudal vena cava to the right heart, leading to venous congestion and edema [3]. The degree of venous inflow obstruction varies depending on the CTD subtype and morphological characteristics of the membrane [4]. CTD can cause various clinical signs in dogs. Ascites and the resulting abdominal distention are the most common signs, as reported in the previously published literature, followed by respiratory distress, pleural effusion, exercise intolerance, stunted growth, and cyanosis [5,6]. Although uncommon, asymptomatic cases have been reported [7,8]. Typically, medical management alone is insufficient to alleviate clinical signs related to CTD [9]; thus, surgical treatment should be considered for dogs with obvious clinical signs associated with CTD. Several methods, such as transcatheter intervention or surgical resection of the persistent membrane via open thoracotomy, have been reported to treat CTD in veterinary medicine [5,10,11,12,13]. This case report describes the clinical signs, diagnosis, and membranectomy via partial venous inflow occlusion (VIO) under mild hypothermia in a dog presenting with clinical signs associated with CTD.

## 2. Case Presentation

A 7-month-old intact male mixed-breed dog weighing 20.5 kg was referred to the Veterinary Medical Teaching Hospital of Chungnam National University for abdominal distention and ascites that were unresponsive to repeated abdominocentesis and diuretic therapy (furosemide; 2 mg/kg, PO, BID) at a local veterinary clinic. The patient has exhibited severe abdominal distention, lethargy, and anorexia since being rescued 3 weeks prior. Physical examination showed within-normal-range vital signs, encompassing heart rate, femoral pulse, cardiac rhythm, and mucous membrane color. Nonetheless, abdominal distention and fluid thrill were observed. Additionally, the patient’s body condition score (BCS) was determined to be 3 out of 9. No other physical exam abnormalities were observed, such as respiratory distress, cardiac murmur, or hepatojugular reflex. Mild anemia (hematocrit, 34.2%; range, 37.3–61.7%), hypoproteinemia (total protein, 4.5 g/dL; range, 5.2–8.2 g/dL), and hypoalbuminemia (albumin, 2.1 g/dL; range, 2.3–4 g/dL) were observed in the blood analysis. The peritoneal fluid analysis revealed a high-protein transudate with a total nucleated cell count of 1350/uL and a total protein (TP) level of 2.6 mg/dL.

Abdominal radiographs showed severe abdominal distention and a loss of serosal detail (Figure 1); no significant findings were observed apart from these. Transthoracic echocardiography showed an echo-dense, band-like structure within the right atrial chamber, extending from the atrioventricular junction to the free atrial wall (Figure 2A). The right atrium was divided into cranial and caudal chambers. Color Doppler imaging revealed continuous venous blood inflow from the caudal chamber to the cranial chamber through a perforation in the remnant membrane, confirming an abnormal venous blood inflow in the right atrium. An agitated saline study was performed using 3 mL of normal saline to further characterize the flow in the right atrium. Microbubbles were made with saline solution, two syringes, and a three-way stopcock. These were injected in the left cephalic vein and the contrast was noted only in the right atrial cranial chamber and the right ventricle immediately afterward (Figure 2B). When the microbubbles were injected in the left lateral saphenous vein, these reached immediately the right atrial caudal chamber; and some moved into the cranial chamber through a perforated membrane (Figure 2C). This confirmed the presence of a cranial true chamber (CrTC) and cranial vena cava (CrVC) as well as a caudal accessory chamber (CdAC) and caudal vena cava (CdVC). A thoracic computed tomography scan was performed using 30 mL of iohexol contrast agent (Omnipaque, GE Healthcare China). Strong contrast enhancement was observed in the cranial chamber that communicated with the CrVC (Figure 3) during the dextrophase (25 s after contrast injection); in contrast, weak contrast enhancement was observed in the caudal chamber connected to the CdVC. A venous blood inflow obstruction was observed in the right atrium connected to the CdVC, suspected of causing severe ascites. Based on these imaging findings, the patient was diagnosed with CTD, and surgical correction was advised. Both interventional procedures and open-heart surgery can be options for treating CTD. However, considering the potential risk of re-stenosis following interventional treatment, a curative resection involving surgical membranectomy via VIO was planned.

The patient underwent preoxygenation with 100% oxygen via the flow-by technique. Preanesthetic medication consisted of diphenhydramine (2 mg/kg, SC; Diphenhydramine HCl Inj., Mylan Ireland), dexamethasone (0.2 mg/kg, IV; Dexamethasone Inj., Jeil ROK), maropitant (1 mg/kg, SC; Cerenia, Zoetis Spain), and cefazolin sodium (22 mg/kg, IV; Cefazolin, Jonggeundang ROK). Propofol (1 mg/kg, IV; Anepol, Hana ROK) and midazolam (0.2 mg/kg, IV; Midazolam Inj., Bukwang ROK) were administered slowly, followed by propofol (4 mg/kg, IV; Anepol, Hana ROK). Anesthesia was maintained using isoflurane (Ifran, Hana ROK). For analgesia, a mixture of lidocaine and bupivacaine (0.2 mg/kg, each; Lidocaine Inj., Jeil ROK; Bupivacaine HCl Inj., Myungmoon ROK) was injected into the intercostal space for intercostal block, and a constant-rate infusion of fentanyl (5 mcg/kg/min; Fentanyl Citrate Inj., Myungmoon ROK) was subsequently administered. When an adequate depth of anesthesia was achieved, the patient was positioned in left lateral recumbency, with the right hind limb reflected for cannulation of the arterial line. The forelimbs were secured in an extended position to prevent limb movements associated with spasms of the extrinsic musculature during thoracotomy. First, the femoral triangle was accessed, and a 5-fr cannula was inserted into the femoral artery using the modified Seldinger technique to measure invasive blood pressure (IBP) in real-time. Subsequently, a right-fifth intercostal thoracotomy was performed using a cardiac approach. For VIO, Rommel tourniquets were placed around the CrVC and CdVC (Figure 4A). In order to facilitate the right atriotomy, a ventral incision of the fibrous pericardium below the phrenic nerve followed by pericardial tenting was performed using 2-0 silk sutures.

Heparin (100 IU/kg) was administered intravenously, and to induce mild hypothermia with a target body temperature of approximately 34.5 °C in the patients, the thermally controlled pad and fluid warmer, which had been used to prevent unintended hypothermia, were discontinued. It took approximately 10 min for the body temperature to decrease to 34.5 °C. Subsequently, a partial VIO was initiated by tightening the tourniquets placed on the CrVC and CdVC. After atriotomy, the cardiac interior was carefully inspected, and the remnant membrane was excised using Metzenbaum scissors (Figure 4B). A vascular clamp was applied to the atriotomy site, and the partial VIO was discontinued. The vascular clamp was then slightly released to allow for de-airing of the atrium, confirming the spillage of a small amount of venous flow and subsequent re-clamping (Figure 4C). The atriotomy site was closed using a two-layer simple continuous pattern with a 6–0 polypropylene suture (Figure 4D). The total duration of partial VIO was 3 min and 30 s, and vital signs returned to normal shortly after the cessation of VIO. No specific abnormalities were observed on arterial blood gas analysis. After completing the intrathoracic procedure, the thoracic cavity was lavaged with warm saline and inspected for air leakage or hemorrhage. To increase the patient’s body temperature, the thermally controlled pad and the fluid warmer were reapplied. A thoracostomy tube was placed intercostally and the surgical site was closed routinely. The arterial line was removed and the femoral artery closed using a simple interrupted pattern with a 6–0 polypropylene suture.

Four days after the procedure, the patient showed normal demeanor and appetite, with no residual ascites (Figure 5A,B). On echocardiography, on day 6 after surgery, normal right atrial flow was detected with no residual obstruction between the cranial and caudal right atrial chambers (Figure 5C,D). There were no specific complications, such as re-stenosis or recurrence of clinical signs related to CTD at the 1-year postoperative follow-up.

## 3. Discussion

Cor triatriatum dexter is characterized by a right atrium that is divided into two compartments; this is due to the presence of a persistent membrane that acts as a septum between the cranial and caudal chambers [4]. As most dogs with CTD have ascites due to the blockage of venous blood flow into the right side of the heart, primarily from the CdVC entering the caudal accessory chamber, abdominal distention is commonly observed on physical examination [6,9,11,12]. Common abdominal radiographic findings include decreased serosal detail due to ascites and an enlarged CdVC. However, clinical signs and physical examination findings may vary depending on the CTD type. Partington et al. [4] reported the classification of CTD into four types based on the vascular structure opening into the right atrium. Type I CTD opens the CrVC, the CdVC, and coronary sinus (CS) into the caudal chamber. Type II CTD opens the CrVC and the CdVC into the caudal chamber, while the CS opens into the cranial chamber. Type III CTD opens the CrVC and the CS into the cranial chamber, while the CdVC opens into the caudal chamber. Type IV CTD opens the CrVC and the CdVC into the cranial chamber, while the CS opens into the caudal chamber. According to previous studies, type III is presumed to be the most common in dogs [5,6,11,12], and there is a single case of type II [4]. In dogs with type II CTD, jugular vein distention and pulsation may be observed as well as hepatojugular reflux [4]. Therefore, it is important to understand the anatomical structures accurately before deciding on surgical correction.

The final diagnosis of CTD is typically made by confirming the division of the right atrium into two chambers by a persistent membrane using transthoracic echocardiography. In addition, the agitated saline bubble test helps assess the pattern of venous inflow and determines the anatomical orientation in dogs with CTD [8]. Contrast echocardiography was performed in our case by injecting agitated saline microbubbles intravenously into the left cephalic vein and the left lateral saphenous vein. In this case, the blood from the CrVC entered the true cranial chamber to cross the tricuspid valve and reach the right ventricle. The CdVC instead opened in the caudal chamber; the flow between the caudal and cranial chamber was obstructed by the presence of a persistent membrane leading to increased CdVC pressure and the onset of the aforementioned clinical signs. Considering the above findings, we diagnosed our patient with type III CTD, as per the classification by Partington et al. [4]. Agitated saline contrast echocardiography is noninvasive, can be performed without general anesthesia, and can provide anatomic and hemodynamic information to establish a treatment plan.

Intravascular or hybrid interventions [5,13], as well as surgical correction through membranectomy via cardiopulmonary bypass (CPB) [10] or VIO [11,12], have been reported as therapeutic methods in dogs with CTD. Interventional techniques have the advantage of being less invasive than open-heart surgery. However, performing dilation precisely at the correct location may be technically challenging if the defect in the persistent membrane is small. There is a possibility of iatrogenic damage to cardiovascular structures, which could result in complications such as hemorrhage, embolism, or arrhythmias [9]. Furthermore, several cases have been reported where additional procedures were required because of re-stenosis after interventional dilation [14,15].

To date, only one case of CTD treatment with CPB has been reported in veterinary medicine [10]. Under CPB, sufficient time can be obtained to inspect the remnant membrane, making it easier for surgeons to perform a membranectomy. However, CPB has a higher risk of critical postoperative complications than VIO, such as systemic inflammatory response syndrome, hemorrhage, low cardiac output, hypotension, arrhythmias, hypoventilation, hypoxemia, decreased urine output, and acid–base and electrolyte abnormalities [16,17]. Furthermore, CPB is more invasive than VIO, prolongs the surgical time, and increases the financial burden on patient owners. As the surgical treatment of CTD requires a very short operation time compared to other open-heart surgeries, unless it is unavoidable, the advantage of CPB are marginal over VIO.

Venous inflow occlusion is a technique that blocks the blood inflow from the CrVC, CdVC, and azygos vein from entering the right atrium [1,16]. By using VIO for CTD surgery, it is possible to minimize postoperative adverse effects, such as hematologic, metabolic, and cardiopulmonary complications, compared with CPB [18], while also enabling direct visual observation of the cardiac interior and facilitating rapid and definitive resection of the obstructive structure. Generally, in normothermic dogs, VIO should be performed within approximately 2 min [1,16]. However, the duration of the obstruction could be extended in hypothermic patients, and Chanoit et al. reported successful inflow occlusion for membranectomy lasting 7 min in a dog under moderate hypothermic conditions (32 °C) [19]. However, performing VIO at a temperature below 32 °C should be cautiously approached due to the significantly higher risk of ventricular fibrillation [1,16]. In our case, VIO was performed for 4 min without adverse effects under mild hypothermia (34.5 °C). This method provided sufficient time for exploration and complete resection of the remnant membrane.

Ventilation is typically suspended during VIO to prevent the influx of pulmonary blood into the surgical field caused by positive pressure ventilation [1]. Following VIO, there is the potential for an increase in end-tidal carbon dioxide (EtCO_2_) levels, which may be attributed to inadequate pulmonary circulation. In this case, intentional hyperventilation was performed before inflow occlusion. Moreover, unlike conventional VIO, blood inflow from the azygos vein was left unobstructed, allowing continuous blood flow into the right ventricle without disrupting the surgical field. It is hypothesized that maintaining inflow from the azygos vein prevents a sudden surge in EtCO_2_ levels upon occlusion release. Furthermore, it is crucial to minimize air entry into the pulmonary circulation and perform de-airing after completion of VIO to prevent pulmonary embolism [16]. Through this partial VIO, which allows some blood flow into the right ventricle, and by performing de-airing prior to complete atrial closure, air entry into the pulmonary circulation is expected to be minimized during the procedure, preventing pulmonary embolism.

## 4. Conclusions

This case report illustrates a successful membranectomy via partial venous inflow occlusion under mild hypothermia for treating a dog diagnosed with type III CTD. By performing partial VIO under mild hypothermia (34.5 °C), there was sufficient time to explore the right atrial chamber and perform the membranectomy. The patient recovered without complications, and the clinical signs associated with CTD resolved. This technique is feasible to treat dogs affected by type III CTD.

## Figures and Tables

**Figure 1 animals-13-02921-f001:**
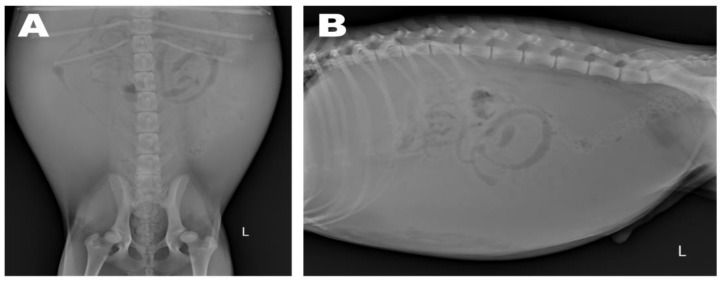
Preoperative abdominal radiography: ventrodorsal (**A**) and lateral (**B**) views. Severe abdominal distention and decreased serosal detail were observed due to ascites.

**Figure 2 animals-13-02921-f002:**
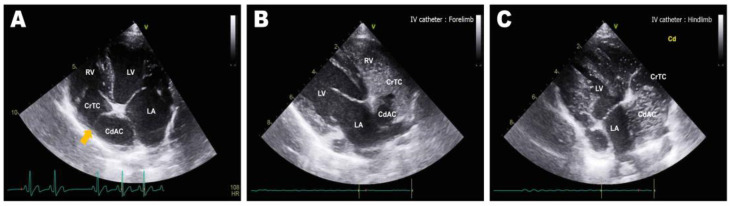
Preoperative transthoracic echocardiography image: left parasternal apical 4-chamber view (**A**), and right parasternal long-axis 4-chamber view tilted to optimize the view of the right atrium (**B**,**C**). (**A**) Septum-like structure (orange arrow) dividing the right atrium into the cranial true chamber (CrTC) and the caudal accessory chamber (CdAC). (**B**) Enhancement of the CrTC and right ventricle (RV) via agitated saline microbubbles after intravenous administration into the cephalic vein. (**C**) Enhancement of the CdAC via agitated saline microbubbles after intravenous administration into the lateral saphenous vein.

**Figure 3 animals-13-02921-f003:**
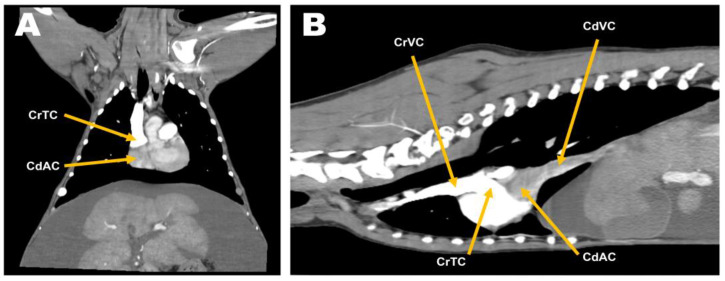
Preoperative computed tomography in the dextrophase: dorsal plane (**A**) and sagittal plane (**B**). While the caudal accessory chamber (CdAC) communicating with the caudal vena cava (CdVC) exhibited no contrast enhancement, the cranial true chamber (CrTC) communicating with the cranial vena cava (CrVC) showed pronounced contrast enhancement. The septum-like structure distinctly divided the right atrium into two distinct sections.

**Figure 4 animals-13-02921-f004:**
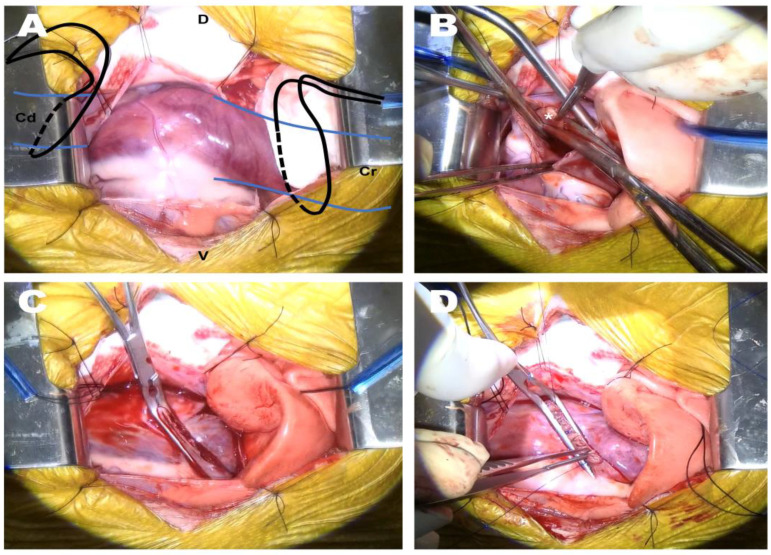
Surgical procedure of partial venous inflow occlusion for membranectomy. (**A**) Preplacement of the Rommel tourniquets (black line) around the cranial and caudal vena cava (blue line). (**B**) Excision of the remnant membrane after right atriotomy. (**C**) Re-clamping of the vascular clamp after de-airing of the right atrium. (**D**) Closure of atriotomy site using two-layer simple continuous pattern. Cr, cranial; Cd, caudal; D, dorsal; V, ventral.

**Figure 5 animals-13-02921-f005:**
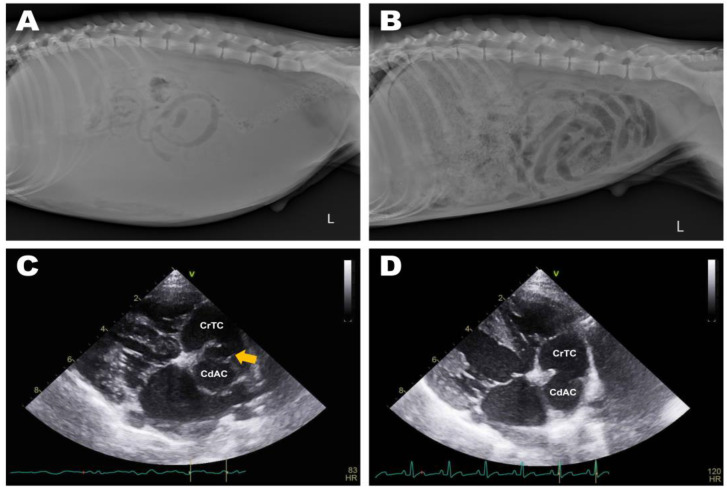
Comparison of preoperative and postoperative diagnostic imaging. Severe abdominal distention and decreased serosa detail (**A**) improved at postoperative day 4 (**B**). The remnant membrane ((**C**), orange arrow) obstructing venous blood inflow from the caudal vena cava was successfully excised (**D**), and blood inflow was not disturbed.

## Data Availability

Data available on request due to restrictions, e.g., privacy or ethical.

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
