# Peer review of "Partial Venous Inflow Occlusion under Mild Hypothermia for Membranectomy in a Dog with Cor Triatriatum Dexter"

_animals, 2023, doi:10.3390/ani13182921_

Round 1
Reviewer 1 Report
The MS “Membranectomy for cor Triatriatum Dexer via milt Hypothermic Inflow Occlusion beating Heart Technique in a Dog”, present for special issue Advances in Animals Cardiac Flysiology aimed to describe the success ful membrane for CTD 7-month-old intact male mixed-breed dog (20,5kg), presenting abdominal distention, le thargy and anorexia.
After many exams, and clinical diagnostic, the authors used the membranectomy to promote partial venous inflow occlusion, followed by a curative resection of cor triatriatum dexter in dogs.
According my oppinion it is an excellent clinical report, however, not enough to support a MS to be published in special issue of Animals.
The MS “Membranectomy for cor Triatriatum Dexer via milt Hypothermic Inflow Occlusion beating Heart Technique in a Dog”, present for special issue Advances in Animals Cardiac Flysiology aimed to describe the success ful membrane for CTD 7-month-old intact male mixed-breed dog (20,5kg), presenting abdominal distention, le thargy and anorexia.
After many exams, and clinical diagnostic, the authors used the membranectomy to promote partial venous inflow occlusion, followed by a curative resection of cor triatriatum dexter in dogs.
According my oppinion it is an excellent clinical report, however, not enough to support a MS to be published in special issue of Animals.
Author Response
We would like to express our gratitude to the editor and reviewers for taking the time to carefully read and to consider the report on our present study. We appreciate the opportunity to revise the manuscript by addressing the reviewers’ comments based on their constructive guidance to help you and the reviewers in your final decision. Below, we have provided detailed and concise point-by-point responses to each of the reviewer concerns in conjunction with the revised manuscript. Our responses in this document and the corresponding changes in the manuscript are written in red.

Reviewer 2 Report
Interesting case with a rare surgical approach and what seems a good outcome! There are several changes that are required to make the manuscript easier to read and understand.
Line 46: “ distal to the obstruction”. Can be erased, is confusing.
Line 48: “clinical symptoms”. Change all to “clinical signs”.
Line 64: “furosemide; 2 mg/kg, PO)”. Which frequency? 2 mg/kg/day? 2 mg/kg PO twice a day?
Line 65: “3 weeks ago.”. Not clear. Three weeks before presentation to the local vet clinic? Before presentation to your hospital?
Line 66: “Furthermore…”. Change to “Physical examination showed..” and list all findings of the complete physical examination including heart rate, femoral pulses, rhythm, mucous membranes, the presence of abdominal distension, the presence of fluid thrill, etc.
Line 76: Typo “ventrodoral”. Ventrodorsal.
Line 77 and 78: Type “serosa”. Serosal.
Line 78: change to “abdominal radiographs showed a loss of serosal details…” and describe the rads.
Line 79: “Transthoracic echocardiography showed an echo-dense, band-like structure within the right atrial chamber, extending from the atrioventricular junction to the free atrial wall.” Were there any other congenital defects? Was the rest of the echo normal?
Line 81: “Owing… chambers.” Not clear. Maybe “the right atrium was divided into a cranial and a caudal chamber. Colour Doppler imaging…”
Line 84-90: “In order to…perforated membrane” Change to “A microbubble study was performed by using X mls of crystalloid/colloid solution injected in the right/left cephalic vein. This resulted in contrast noted only in the cranial chamber and in the right ventricle immediately afterwards. Eight minutes after, bubbles were observed in the caudal chamber with minimal amount of contrast crossing the septum-like structure to the cranial chamber. When the microbubbles were injected in the left/right lateral saphenous vein, these reached immediately the right atrial caudal chamber. This confirmed the presence of a true cranial chamber and cranial vena cava as well as a caudal chamber and caudal vena cava.”
Line 88: What is the proposed explanation for the 8-min delay of appearance of contrast in the caudal chamber?
Line 93: “During the early arterial…”. Here we are talking about a CT exam and there is no mention that this was performed leaving the reader confused. Start with “A thoracic CT scna was performed using X mls of X contrast agent injected through the X vein.”
Line 98: “type III CTD”. Mention which reference you used for this classification.
Line 98: change to “surgical correction was advised”.
Line 98: expand on which other corrective options are possible.
Line 100: “via VIO was planned” , remove “the”.
Line 103: “Perioperative transthoracic echocardiography image” (add “image”)
Line 103: figure 2. It doesn’t help understanding to have the first image standardized and the other flipped R to L to optimize the right heart. Make them all with same orientation.
Line 109: Figure 3. The arterial phase shows contrast in the cranial vena cava and right heart but not in the left heart or aorta. Is this an arterial phase? Should probably be called “dextrophase”.
Line 114: all drugs should have a the commercial name, followed by manufacturer i.e. “cefuroxime sodium (Zinacef, GlaxoSmithKline UK)” unless not-required by the editors.
Line 132: How long did you have to wait for temperature to reacah 34.5. Why were fluid warmer used in the first instance if we were aiming to induce hypothermia?
Line 153: You have removed the arterial line but you did not close the arterial line. Change to ” the arterial line was removed and the femoral artery closed using a simple interrupted..”
Line 155: Change to “Four days after the procedure the patient showed normal demeanour and appetite, with no residual ascites.
Line 157: why are we mentioning here that the remnant membrane was excised?
Line 158: change to “On echocardiography, on day X after surgery, normal right atrial flow was detected with no residual obstruction between cranial and caudal right atrial chambers”
Line 158: How did you confirm there was no more obstruction via echocardiography? Did you use Colour Doppler? Or simply in 2D images?
Line 169: this sentence starts in a awkward way. Maybe “Cor triatratum dexter is characterized by a right atrium that is divided into…. This is due to the presence of a persistent membrane that act as a septum between the cranial and caudal chambers”.
Line 173: “common abdominal radiographic findings…”
Line 175: describe which are the subtypes and which reference you used.
Line 177: “instead of” or “as well as” ?
Line 185: “specifically targeting …” remove. Change to “injecting agitated saline microbubbles into the cephalic (left or right?) and lateral saphenous (left or right?) vein”.
Line 186: poor clarity. “In this case, the blood entered from the CrVC to the true cranial chamber to the cross the tricuspid valve and reach the right ventricle. The CdVC instead opens in the caudal chamber; the flow between the caudal and cranial chamber is obstructed by the presence of a persistent membrane leading to increased CdVC pressure and the onset of the aforementioned clinical signs”.
Line 188: “Based on… within the right atrium”. Erase.
Line 190: “this led to… “ change to “Considering the above findings, we diagnosed our patient with a type III CTD, as per classification by (author of reference)”.
Line 205: “Therefore,… dogs with CTD.” Erase
Line 215: “VIO…” don’t start a sentence with an acronym but use full word. Remove the brackets from the vessels. Remove “allowing intracardiac…quickly”.
Line 218: “heamatologic…abnormalities”. Change to complications.
Line 222: I am not sure “circulatory arrest” is the correct definition. Can be removed and leave “the duration of the obstruction..”
Line 234: rephrase and explain why the Azigos flow would not result in right atrial bleeding. Also, if you talk about this case, use the past tense. I.e. blood inflow from the azygos was not occluded.
Line 236: there is no mention of “de-airing” when describing your procedure.
Line 242: “using a mild” not “using the mild”
Line 243: “for treating a dog diagnosed with type III CTD”.
Line 244: “By performing partial VIO under mild hypothermia (34.5 ℃), there was sufficient time to explore the right atrial chamber and perform the membranectomy”.
Line 246 “and clinical signs associated with CTD resolved.”
Line 247: “This technique is feasible to treat dogs affected by type III CTD”.
The reference list is not standardized. Some have DOI, some haven’t.
Generally, make sure the investigations process, findings and therapy solutions and approaches are discussed in a chronologically logic order. In the discussion follow the same process.
Make sure all sentences are in correct English, with the same verb tense and follow a logic approach to the case.
Author Response

(The authors gave the same response as above.)

Reviewer 3 Report
Congratulations on your very interesting case report, well done! Laurent Locquet

See notes
Author Response

(The authors gave the same response as above.)

Reviewer 4 Report
Thank you to the authors for this interesting insight as it does shed additional light onto treatment options for CTD. I have mostly minor comments to make:
Major comment:
Since Type III CTD was brought up, would a quick review of the different types be useful? I believe the other types are very rare (if even reported).
Minor comments:
Line 41: Delete “The” at the beginning of the sentence.
Line 45: Change “vena cavae” to “caudal vena cava”.
Lines 48-49: “symptom” refer to what people are feeling. Since we cannot discern that in dogs, you can change it to “clinical signs and physical exam findings” (cardiac murmur is neither a clinical sign or a symptom). That being said, you can consider removing cardiac murmur because murmurs are not expected in a typical CTD.
Line 67: Change “clinical symptoms” to “physical exam abnormalities”.
Line 72: No need to abbreviate since these terms are no longer referenced in the remainder of the text.
Lines 78-100: Font size should be returned to normal.
Line 78: Can delete “other”.
Line 81: Change “its” to “this”.
Line 82: Consider inserting “continuous” before “venous blood inflow”.
Line 84: Consider changing “into the right heart” to “in the right atrium”.
Lines 84-85: Consider just simply stating: “An agitated saline study was performed using two syringes and a 3-way stopcock to further characterize flow in the right atrium”.
Line 87: Can delete “initially” as the bubbles probably never entered into the caudal chamber. On that note, I find it hard to believe that you watched the bubbles for 8 minutes. Unless a concurrent lesion is present, the bubbles should be cleared by the lungs and not enter the left side of the heart, let alone return to the caudal right atrium.
Line 93: Best to state that this diagnostic test is computed tomography, since when first reading this sounds like it is still the agitated saline study.
Figure 2A: specify left apical 4-chamber view
Figure 2B+2C: specify which view as this is different than in 2A
Figure 2: Define LV and LA
Lines 110-112: Please rephrase and this sentence is confusing and hard to follow
Line 116-117: Was the entire 4 mg/kg propofol given or was it titrated to effect?
Lines 117-118: Why was a mixture of lidocaine given and not lidocaine or bupivacaine alone? I’m not an anesthesiologist so I’m just curious why a mixture was used.
Line 129: “followed by” rather than “following”?
Line 167: Change “no” to “not”.
Line 174: “Clinical signs” rather than symptoms
Line 183: Consider changing “structure” to “orientation”
Line 201: Change “where” to “that”
Line 202: Change “are” to “were”
Lines 213-214: Consider saying “the advantages of CPB are marginal over VIO” to make the sentence more streamlined.
Line 216: Add “from” before “entering”
Line 234: Consider changing “supply” to “flow”
Suggestions for edits are included above.
Author Response

(The authors gave the same response as above.)

Round 2
Reviewer 1 Report
I agree with the changes
I agree with the changes
Author Response
We would like to express our gratitude to the editor and reviewers for taking the time to carefully read and to consider the report on our present study. We appreciate the opportunity to revise the manuscript by addressing the reviewers’ comments based on their constructive guidance to help you and the reviewers in your final decision. Below, we have provided detailed and concise point-by-point responses to each of the reviewer concerns in conjunction with the revised manuscript. Our responses in this document and the corresponding changes in the manuscript are written in red.
For further details, we will attach a Word file formatted as provided by you.

Reviewer 2 Report
Line 81. "The right atrium is ...". Change to "was" , keep the same tense, past in this case.
Line 86. I appreciate this has been changed but still is not clear and the use of verbs is inappropriate as well as plurals. "After injecting microbubble made by two syringe and a 3-way stopcock in the left cephalic vein, resulted contrast noted only in the right atrial cranial chamber and the right ventricle immediately afterwards." Change to "Microbubbles were made with saline solution, two syringes and a 3-way stopcock. These were injected in the left cephalic vein and contrast was noted only in the 87 right atrial cranial chamber and the right ventricle immediately afterwards."
Line 140: How long did you have to wait for temperature to reacah 34.5. Why were fluid warmer used in the first instance if we were aiming to induce hypothermia?. I have seen the answer on the .docx but this was not answered or added in the text. Please add the the response to the text.
Line 191. "in dog with.." change to "in dogs", plural.
Line 199. "by intravenously injecting agitated saline microbubbles" change to "by injecting agitated saline microbubbles intravenously".
Line 200. "In this case, the blood entered from the CrVC to the true cranial chamber" . Change to "In this case, the blood from the CrVC entered the true cranial chamber..."
Line 202. "The CdVC instead opens.." Change to "opened". Keep using the past tense. Line 204. change to "was obsteructed"
Line 259. "the right atrial chamber AND perform the membranectomy".
There are still some imprecisions. Past/presence tense change and plurals. Review this.
Author Response

(The authors gave the same response as above.)
